# Letrozole at the Crossroads of Efficacy and Fetal Safety in Ovulation Induction: A Narrative Review

**DOI:** 10.3390/biomedicines13092051

**Published:** 2025-08-22

**Authors:** Aris Kaltsas, Anna Efthimiou, Christos Roidos, Vasileios Tzikoulis, Ioannis Georgiou, Alexandros Sotiriadis, Athanasios Zachariou, Michael Chrisofos, Nikolaos Sofikitis, Fotios Dimitriadis

**Affiliations:** 1Third Department of Urology, Attikon University Hospital, School of Medicine, National and Kapodistrian University of Athens, 12462 Athens, Greece; ares-kaltsas@hotmail.com (A.K.); mchrysof@med.uoa.gr (M.C.); 2Department of Obstetrics, Gynecology and Assisted Reproduction, Maternity—Health, Chalandri, 15232 Athens, Greece; anna.efthim@gmail.com; 3Department of Urology, Faculty of Medicine, School of Health Sciences, Aristotle University of Thessaloniki, 54124 Thessaloniki, Greece; drchriroid22@gmail.com (C.R.); bill1996tziko@gmail.com (V.T.); 4Laboratory of Medical Genetics in Clinical Practice, Faculty of Medicine, School of Health Sciences, University of Ioannina, 45110 Ioannina, Greece; igeorgio@uoi.gr; 5First Department of Obstetrics and Gynecology, Papageorgiou General Hospital of Thessaloniki, Faculty of Medicine, School of Health Sciences, Aristotle University of Thessaloniki, 56403 Thessaloniki, Greece; asotir@gmail.com; 6Laboratory of Spermatology, Department of Urology, Faculty of Medicine, School of Health Sciences, University of Ioannina, 45110 Ioannina, Greece; azachariou@uoi.gr (A.Z.); nsofikit@uoi.gr (N.S.)

**Keywords:** letrozole, ovulation induction, polycystic ovary syndrome, clomiphene resistance, infertility management, ovarian hyperstimulation, estrogen-sensitive cancers

## Abstract

Letrozole, a third-generation aromatase inhibitor initially developed for breast cancer, has become the preferred first-line agent for ovulation induction (OI), particularly in women with polycystic ovary syndrome (PCOS). This narrative review critically evaluates the efficacy, safety, and clinical applications of letrozole across diverse infertility contexts. Compared to clomiphene citrate, letrozole is associated with higher ovulation and live birth rates, a lower risk of multiple gestation, and a more favorable endometrial environment. Its pharmacokinetics—marked by transient estrogen suppression and a short half-life—limit embryonic exposure, supporting its favorable safety profile. Emerging data from large, randomized trials and meta-analyses demonstrate no increase in congenital anomalies, miscarriage, or adverse perinatal outcomes in letrozole-conceived pregnancies. Moreover, maternal side effects are generally mild, and the risk of ovarian hyperstimulation syndrome is low. Letrozole has also shown utility in mild stimulation protocols, fertility preservation for estrogen-sensitive malignancies, and clomiphene-resistant PCOS. Key clinical strategies—such as early-cycle initiation, lowest effective dosing, and individualized monitoring—optimize therapeutic outcomes while minimizing potential risks. While long-term offspring data remain limited and mechanistic concerns persist, current evidence robustly supports letrozole as a safe and effective option for OI, balancing reproductive success with maternal–fetal safety across a range of infertility indications.

## 1. Introduction

Infertility is defined as the failure to achieve a pregnancy after 12 months of regular unprotected intercourse [1]. It is a complex condition with multifactorial causes and affects approximately 10–15% of couples of reproductive age worldwide [2,3]. Both female and male factors contribute: female infertility factors (e.g., ovulatory dysfunction, tubal disease, endometriosis) account for roughly 35–50% of infertility cases, often overlapping with male factors, and male factors are involved in 40–50% [4,5]. Ovulatory disorders— such as the chronic anovulation in polycystic ovary syndrome (PCOS)—are a common cause of female infertility and a primary target for medical therapy [6]. Mounting evidence indicates that the low-grade inflammation and oxidative-stress milieu characteristic of PCOS further impairs oocyte quality and increases miscarriage risk [7].

Ovulation induction with pharmacotherapy is a cornerstone of infertility treatment for anovulatory women [8]. Letrozole, a third-generation aromatase inhibitor initially developed for treating breast cancer, has emerged as a first-line OI agent, particularly in anovulatory PCOS [9]. Current clinical practice guidelines, such as those from the American Society for Reproductive Medicine (ASRM) and the European Society of Human Reproduction and Embryology (ESHRE), recommend letrozole as the preferred first-line pharmacological treatment for inducing ovulation in infertile women with anovulatory PCOS, given its superior efficacy and safety profile compared to older agents like clomiphene citrate [6,10]. Additionally, the international evidence-based guideline spearheaded by the international PCOS network also endorses letrozole as the first-line ovulation induction therapy for anovulatory PCOS [6]. Letrozole’s growing use in infertility care reflects its higher ovulation and live birth rates (especially in PCOS) and its lower risk of multiple pregnancy relative to clomiphene [11,12]. The anti-estrogenic adverse effects of clomiphene on cervical mucus and endometrium are avoided with letrozole, resulting in a more favorable endometrial lining and cervical mucus environment for conception [13]. Letrozole is also used in mild ovarian stimulation protocols and in fertility preservation for hormone-sensitive cancers due to its transient estrogen-lowering action [14].

Although letrozole is used off-label for infertility in many countries, early concerns about teratogenicity led to its categorization as a pregnancy category X drug based on its oncological use. Subsequent observational reports and case series suggested a possible association with congenital anomalies, but these were later discredited due to significant methodological flaws, including small sample sizes and lack of comparison groups [15,16]. A 2008 review emphasized that letrozole did not increase birth defect risk and noted that clomiphene citrate had more established associations with adverse fetal outcomes [16]. Despite being supported by solid clinical evidence, these early alarms continue to influence perceptions among some clinicians and patients regarding letrozole’s safety when used for conception.

Over the past decade, large randomized trials and meta-analyses have largely quelled safety concerns, demonstrating no significant increase in birth defects or pregnancy loss associated with letrozole compared to other fertility treatments [17,18]. However, data are still limited on long-term offspring health beyond the neonatal period, and the mechanistic understanding of how transient estrogen suppression affects early embryogenesis remains an evolving field [19]. Therefore, while the benefits of letrozole for infertility treatment are clear, it is crucial to balance them thoughtfully with the best available safety evidence for both mother and fetus.

The aim of this narrative review is to critically evaluate the efficacy of letrozole for ovulation induction, assess its fetal safety profile, and provide clinically relevant guidance for its application in reproductive medicine. Current clinical guidelines and high-quality evidence are synthesized to elucidate the pharmacological mechanisms, therapeutic efficacy, and safety of letrozole across various clinical scenarios, including PCOS, unexplained infertility, combination protocols, and special populations. Data from preclinical models and mechanistic studies concerning potential teratogenic pathways are also examined. Strategies to mitigate risk are outlined, with emphasis on optimizing patient outcomes and minimizing theoretical fetal exposure. This review aims to support evidence-based clinical decision-making regarding the use of letrozole in infertility treatment and to identify areas requiring further investigation.

## 2. Pharmacology of Letrozole

### 2.1. Mechanisms of Action

Letrozole is a non-steroidal, third-generation aromatase inhibitor that competitively blocks the cytochrome P450 19A1 (CYP19A1) enzyme, preventing conversion of androgens (androstenedione, testosterone) to estrogens (estrone, estradiol) [20]. Short-term administration during the early follicular phase suppresses ovarian and peripheral estrogen synthesis by approximately 97–99%, producing a transient hypo-estrogenic milieu [21,22]. The resulting decline in circulating estradiol removes the normal negative feedback on the hypothalamic–pituitary axis, leading to increased pulsatile gonadotropin-releasing hormone (GnRH) output and a subsequent rise in pituitary follicle-stimulating hormone (FSH) and luteinizing hormone (LH) secretion [23]. Elevated FSH promotes growth of ovarian follicles and can restore mono-ovulation in anovulatory women.

Intra-ovarian aromatase inhibition also causes a temporary accumulation of androgens within the follicular microenvironment. These androgens up-regulate FSH-receptor expression and insulin-like growth factor-1 (IGF-1) production in granulosa cells, thereby amplifying follicular sensitivity to FSH and further enhancing folliculogenesis [24,25]. Once letrozole is discontinued—typically by cycle day 7–9—estradiol levels rise again as the dominant follicle matures, reinstating negative feedback and limiting recruitment to a single dominant follicle [26].

The differential endocrine pathways underlying ovulation induction with letrozole compared to clomiphene citrate are illustrated in Figure 1. While letrozole exerts its effect by reducing estrogen synthesis without interfering with estrogen receptors, clomiphene acts as a selective estrogen receptor modulator (SERM), blocking hypothalamic estrogen receptors and simulating estrogen deficiency. This distinction underpins their divergent clinical and endometrial profiles, despite both enhancing GnRH and gonadotropin secretion.

### 2.2. Pharmacokinetics and Dosing for Infertility

Letrozole exhibits nearly complete oral bioavailability (≈99.9%) and is well-absorbed following administration. Its mean terminal elimination half-life ranges between 41 and 48 h (approximately 2 days), and steady-state concentrations are typically achieved after one week of daily dosing. Metabolism is primarily hepatic—mainly via CYP2A6 (with lesser roles for CYP3A4)—to inactive metabolites that undergo glucuronidation; approximately 90% of letrozole is excreted in the urine, predominantly as conjugates [27].

In ovulation induction protocols, letrozole is administered orally at 2.5 mg daily for five days, commonly initiated from cycle day (CD) 3 to CD 5 of a spontaneous or progesterone-induced menses. Ovulation generally occurs around 7–10 days after the final dose. Dose escalation to 5 mg or 7.5 mg daily—while still limited to five days—may be employed in non-responders, though evidence supporting higher dosing remains moderate [28]. A 2025 systematic review and network meta-analysis of 30 RCTs involving 3663 PCOS patients showed that 5 mg and 7.5 mg letrozole nearly doubled clinical-pregnancy odds versus the 2.5 mg dose, identifying moderate-dose regimens as the most effective strategy [29]. Extended regimens, such as 2.5 mg daily for 7–10 days, have demonstrated equivalent ovulation and pregnancy rates compared with standard dosing in clomiphene-resistant PCOS populations, without increased risk of ovarian hyperstimulation syndrome [30]. Thus, standard infertility dosing ranges from 2.5 to 7.5 mg daily for five days, with extended or higher-dose regimens reserved for cases of initial non-response [31]. To minimize embryonic exposure, letrozole is typically discontinued by CD 7–9; given its 42–48-hour half-life, the drug is effectively cleared prior to ovulation and conception [32]. This practice ensures negligible drug levels by the time of fertilization.

### 2.3. Comparison with Clomiphene and Gonadotropins

Letrozole differs pharmacologically and clinically from both clomiphene citrate and injectable gonadotropins (FSH ± LH). Clomiphene has a long half-life with active metabolites that persist for weeks, often inducing anti-estrogenic carry-over effects into subsequent cycles. In contrast, letrozole’s shorter half-life (~2 days) allows each treatment cycle to remain hormonally independent [17].

Importantly, letrozole does not antagonize estrogen receptors in peripheral tissues, while sparing endometrial development. Randomized controlled trials in women with PCOS have demonstrated significantly greater endometrial thickness with letrozole (mean ~9.16 mm) compared to clomiphene (~4.46 mm), along with improved uterine blood flow [33,34].

Clinically, letrozole increases monofollicular ovulation rates, demonstrates superior efficacy in clomiphene-resistant PCOS populations [35], and results in lower rates of multiple gestations than gonadotropins. In the AMIGOS trial, gonadotropins induced 32% multiple pregnancies—many being higher-order—while letrozole and clomiphene were associated predominantly with twins (≤13%) [36]. Similarly, in intrauterine insemination (IUI) studies, letrozole produced lower twin rates (~4%) than combined letrozole plus HMG protocols (~11%) or gonadotropins alone [37].

Gonadotropins remain the most potent ovulation-induction modality—achieving pregnancy rates of 15–30% per cycle—but carry significant risks: high rates of multifollicular development, multiple gestation, and ovarian hyperstimulation syndrome (OHSS), necessitating intensive monitoring [38]. Letrozole, by eliciting a more physiological FSH surge, typically induces only 1–2 dominant follicles, markedly reducing OHSS and high-order multiple pregnancy risk. Furthermore, side-effect profiles differ: clomiphene is associated with hot flashes, visual symptoms (~1–2%), mood changes, and breast tenderness, whereas letrozole tends to cause only mild fatigue, dizziness, or headaches, and less frequent vasomotor symptoms [39]. Gonadotropins require daily injections, frequent ultrasound monitoring, and incur greater cost and patient burden.

Letrozole exhibits distinct pharmacological properties and clinical characteristics when compared with clomiphene citrate and injectable gonadotropin preparations (FSH ± LH), as summarized in Table 1.

## 3. Clinical Effectiveness of Letrozole in Ovulation Induction

Letrozole is now the preferred first-line oral agent for ovulation induction in many infertility practices, reflecting the robust evidence supporting its use. Its efficacy has been evaluated across various patient populations and in comparison with alternative therapies. Guidelines now reflect these findings: the 2023 International Evidence-based Guideline for PCOS recommends letrozole as first-line pharmacologic therapy for anovulatory infertility [6]. Letrozole is particularly effective in obese PCOS—live-birth rates were roughly doubled versus clomiphene in the BMI > 30 kg/m^2^ subgroup of the Legro trial [40]. 

Unexplained infertility is often managed with empiric ovarian stimulation combined with IUI, and letrozole has emerged as an effective and safer alternative to clomiphene in this context. In the NICHD network trial (Diamond et al., 2015), live-birth rates after up to four IUI cycles were similar between letrozole (18.7%) and clomiphene (23.3%), yet multiple-gestation rates were much lower with letrozole (13% vs. 32% for gonadotropins) [8]. A 2019 meta-analysis found comparable live-birth rates between letrozole and clomiphene (pooled RR ~0.94) with a trend toward fewer twin pregnancies on letrozole [41]. Moreover, a 2025 scoping review of 12 ART studies reported that letrozole-only stimulation achieved the highest clinical pregnancy (~51%) and live-birth (~46%) rates, whereas letrozole-plus protocols yielded the greatest ongoing-pregnancy rate (~58%) with the lowest miscarriage risk (~15%), highlighting the agent’s versatility across infertility subgroups [42].

Combining letrozole with other agents expands its clinical utility, particularly in complex or high-risk cases such as estrogen-sensitive malignancies. In such contexts, co-treatment with letrozole during controlled ovarian stimulation—commonly referred to as COST-LE (Controlled Ovarian Stimulation with Letrozole)—has been shown to lower peak estradiol levels by approximately 70% without compromising oocyte yield [43]. In a cohort study of 120 breast cancer survivors, this strategy did not increase the five-year risk of cancer recurrence (hazard ratio 0.77), supporting its safety for fertility preservation in oncology patients [44]. Beyond oncology, combination oral therapy has also been explored in PCOS: a 2019 RCT (Mejia et al.) reported that letrozole plus clomiphene achieved a 77% ovulation rate versus 43% with letrozole monotherapy in clomiphene-resistant cases, although live-birth data remain limited [45]. A 2025 systematic review and meta-analysis of four RCTs and two observational studies (*n* = 592) confirmed that letrozole + clomiphene significantly increases the likelihood of ≥1 mature follicle (OR 2.74) and ovulation (OR 2.55) versus letrozole alone, while pregnancy rates and safety profiles remain similar [46]. Similarly, hybrid protocols combining letrozole with low-dose FSH in a sequential fashion appear to enhance follicular recruitment while maintaining a mild ovarian response; a 2024 pragmatic RCT showed higher pregnancy rates with such a protocol compared to letrozole alone, without increasing the rate of multiple gestation [47].

Dose extension and resistance management strategies are essential when patients fail to ovulate on standard letrozole regimens, and several approaches have been developed to address this challenge. One commonly used method is extending the duration of letrozole administration to 7–10 days, which can overcome resistance in many non-responders. A 2024 randomized controlled trial comparing a 7-day versus a conventional 5-day regimen showed ovulation rates of 90% versus 80%, respectively (not statistically significant), with similar pregnancy outcomes—supporting the use of extended dosing before increasing the daily dose [48]. Another strategy involves sequential “stair-step” protocols, which escalate the dose within the same cycle without waiting for withdrawal bleeding. These protocols have been shown to shorten time to ovulation and improve patient convenience [49]. In addition, adjunctive use of metformin in women with insulin-resistant PCOS has been shown to enhance ovulatory response and is supported by contemporary PCOS guidelines [6]. If ovulation still fails at the maximum letrozole dose (7.5 mg daily) even with a 10-day course, the next step typically involves low-dose gonadotropin therapy or transitioning to IVF.

## 4. Fetal and Maternal Safety Profile

Letrozole’s safety in the context of infertility treatment has been examined through preclinical studies and extensive clinical data. Early concerns prompted by animal findings and isolated reports have given way to a robust body of evidence supporting a favorable fetal and maternal safety profile. Below, we review the key evidence on preclinical teratogenicity, human congenital anomaly risk, neonatal outcomes, and maternal adverse events including ovarian hyperstimulation considerations.

### 4.1. Preclinical Data 

Initial animal studies raised theoretical safety concerns. In pregnant rats, letrozole exposure during organogenesis led to dose-dependent increases in post-implantation loss and fetal malformations [50]. Teratogenic effects observed included skeletal anomalies (e.g., vertebral malformations and delayed ossification) and other abnormalities such as renal papilla absence and edema [51]. For instance, Tiboni et al. (2008) reported that even low doses (comparable to human exposure) resulted in significantly higher fetal resorptions and minor vertebral defects in rat offspring [52]. Histopathological changes in the rat placenta have also been documented. Furukawa et al. (2024) found that letrozole-treated rats developed multiple placental cysts lined with undifferentiated trophoblast cells by mid-gestation, leading to enlarged, congested placentae [53]. These placental changes were associated with transiently increased fetal weights early in gestation and subsequent intrauterine growth restriction as pregnancy progressed [53]. Such findings underscore the potential for developmental disruption in an estrogen-deprived in utero environment. It is important to note, however, that these effects in animals occurred at doses or exposure periods that may not directly translate to human clinical use. Notably, letrozole’s short half-life (~2 days) means the drug is largely cleared before the phase of human fetal organogenesis [13]. This pharmacokinetic detail suggested that the window of fetal exposure in clinical use would be limited.

In summary, while animal models indicated possible skeletal malformations and placental abnormalities from letrozole, these preclinical signals necessitated careful monitoring but did not conclusively predict human risk. The early animal data prompted caution and further investigation into letrozole’s safety in human pregnancy, as discussed below.

### 4.2. Congenital Anomalies in Humans

Initial case reports and a non-peer-reviewed study in the mid-2000s raised alarm about possible congenital cardiac defects in letrozole-exposed pregnancies, leading some regulators to temporarily contraindicate letrozole in women of childbearing age [54]. However, subsequent rigorous analyses have refuted an association between letrozole and birth defects. Additionally, two large retrospective cohort studies published in the early 2020s bolster the safety data. Takeshima et al. (2021) examined perinatal outcomes and anomalies in over 1000 singleton pregnancies conceived in letrozole cycles (including some IVF-letrozole cycles) compared to those conceived in natural cycles. They found no significant differences in rates of stillbirth, neonatal death, preterm birth, or major congenital anomalies between the letrozole group and either clomiphene or natural conception groups [13]. Similarly, Wang et al. (2024) reported on 194 letrozole-induced singleton pregnancies versus 154 control (non-letrozole) pregnancies and observed no increase in adverse neonatal outcomes [19]. The letrozole group did not have higher odds of preterm delivery, low birth weight, or neonatal intensive care admissions, and the frequency of birth defects was statistically comparable to the control group. Interestingly, Wang et al. noted the cesarean delivery rate was lower in the letrozole group (44% vs. 56%, *p* = 0.019), though the reasons for that are unclear and could relate to obstetric practices [19]. Overall, these human studies collectively indicate that letrozole use for ovulation induction is not associated with an increased risk of major fetal anomalies or poor perinatal outcomes when compared to other standard treatments.

Pundir et al. (2021) conducted a comprehensive systematic review and meta-analysis of the reproductive safety of letrozole. In their meta-analysis of 46 studies including 4697 letrozole-associated pregnancies, the overall rate of congenital malformations in infants conceived with letrozole was 2.15%, which is comparable to background rates [17]. Crucially, letrozole did not confer any statistically significant increase in the risk of major birth defects compared to clomiphene citrate or to natural conception. In pooled randomized trials, the risk difference in congenital anomaly rates between letrozole and clomiphene was essentially zero (RD ~0.0, 95% CI spanning no effect) [17]. Intriguingly, in observational cohort comparisons, letrozole was associated with a slightly lower malformation risk than clomiphene (approximately 2% absolute risk reduction) [17]. Although that finding may reflect some bias or differences in population, it reinforces that no excess teratogenic effect of letrozole has been detected in humans. The same meta-analysis also examined pregnancy losses and found no increase in miscarriage or pregnancy loss rates with letrozole use [17]. In fact, some data suggested that miscarriages were less frequent in letrozole-treated pregnancies than in clomiphene-treated ones in observational settings [17]. 

Overall, the high-quality evidence now available provides reassurance that letrozole does not increase the incidence of congenital anomalies or spontaneous pregnancy loss in humans [17]. These findings have led experts to conclude that prior warnings were not warranted, and that letrozole can be considered a safe first-line ovulation induction agent from a fetal malformation standpoint [17]. In summary, large meta-analyses of recent data conclusively indicate no elevated risk of birth defects in letrozole-conceived offspring compared to alternative treatments or natural conceptions.

### 4.3. Neonatal and Perinatal Outcomes

Beyond congenital anomalies, the perinatal outcomes of letrozole-associated pregnancies have been extensively studied to identify any subtler risks to newborns. Key outcomes such as birth weight, gestational age at delivery, and neonatal health indicators (e.g., Apgar scores and neonatal complications) appear to be similar between letrozole and other fertility treatments. For example, a recent cohort study in China compared 194 singleton births from letrozole ovulation induction to 154 from non-letrozole treatments and found no adverse differences in birth outcomes [19]. Mean birth weights and lengths were equivalent between groups, and the rate of preterm delivery was not increased with letrozole (the proportion of full-term vs. preterm births did not differ significantly) [19]. The incidence of low birth weight and small-for-gestational-age infants was also comparable. Wang et al. (2024) reported that maternal letrozole use was not associated with any increase in neonatal complications, with adjusted analyses confirming no significant impact on neonatal health [19]. Similarly, Kato et al. (2022) analyzed over 2800 births from letrozole-minimal stimulation IVF cycles versus natural cycles and observed no differences in mean gestational age or rates of infants born small or large for gestational age [13]. The letrozole and natural-cycle groups had statistically indistinguishable perinatal profiles, including pregnancy complication rates and newborn outcomes [13]. A 2025 RCT in 200 PCOS patients undergoing frozen-embryo transfer likewise found comparable clinical-pregnancy rates between letrozole-primed and HRT cycles (66% vs. 53%; RR 1.25, *p* = 0.061), while letrozole improved biochemical pregnancy and live-birth rates in normal-weight, normo-androgenic women [55]. These data suggest that letrozole does not predispose to obstetric complications like preterm birth, nor to fetal growth issues such as intrauterine growth restriction, beyond baseline rates. One important factor contributing to favorable neonatal outcomes is the lower incidence of multiple gestation with letrozole. Letrozole tends to promote monofollicular ovulation; accordingly, twin pregnancies are significantly less frequent with letrozole than with clomiphene in comparative studies [40]. This reduction in multiple gestations translates to a lower risk of preterm delivery and low birth weight sequelae, indirectly improving perinatal outcomes. Taken together, the current evidence indicates that infants conceived with letrozole have perinatal outcomes (term birth rates, birth weight distribution, neonatal well-being) that are equivalent to those conceived with other ovulation induction approaches or spontaneous pregnancies [19]. No specific pattern of adverse neonatal outcome has been linked to letrozole use. This consistency in neonatal health measures further reinforces the safety of letrozole from a fetal/newborn perspective, complementing the congenital anomaly data reviewed above.

### 4.4. Maternal Adverse Events and OHSS Risk Mitigation

From the maternal standpoint, letrozole is generally well tolerated and offers certain safety advantages over alternative ovulation induction therapies. Its pharmacologic profile—a short-acting aromatase inhibitor—confers a relatively mild side-effect burden. Clinical trials have noted that letrozole’s side-effect profile is favorable when compared to clomiphene. In the largest randomized trial in polycystic ovary syndrome, clomiphene was associated with significantly more frequent vasomotor symptoms (notably hot flushes), whereas letrozole users reported slightly higher rates of mild fatigue and dizziness [40]. Other side effects (e.g., nausea, headache) were low in incidence and similar between the two drugs [40]. Importantly, because letrozole does not exert an anti-estrogenic action on central estrogen receptors like clomiphene, it has less negative impact on mood and endometrial thickness [40]. Investigators have highlighted that letrozole provides a “more physiologic” hormonal environment for the endometrium, avoiding the persistent estrogen receptor blockade that clomiphene induces [40]. Indeed, few letrozole-related adverse effects on the endometrium have been reported in the literature, and patients on letrozole generally do not experience the anti-estrogenic side effects (such as thin endometrium or notable menopausal symptoms) often seen with clomiphene [56,57]. -Overall, the maternal adverse event profile of letrozole is mild, with most women experiencing no significant side effects; this contributes to letrozole being considered a patient-friendly option [17].

Another critical maternal safety consideration in infertility therapy is the risk of ovarian hyperstimulation syndrome (OHSS). Letrozole-based regimens inherently carry a low risk of OHSS because they stimulate recruitment of only one or a few follicles and keep peak estradiol concentrations well below the thresholds typically reached with gonadotropins [58]. In large randomized trials and meta-analyses of ovulation-induction cycles that used oral letrozole alone, clinically significant or severe OHSS has been extremely rare, with most studies reporting zero cases [39].

In controlled ovarian-stimulation (COS) protocols for high-risk patients—particularly women with PCOS who exhibit high antral-follicle counts or very high serum AMH—letrozole has been evaluated as an adjunct to mitigate hyper-response. Co-administration of letrozole with a GnRH-antagonist protocol significantly reduced the incidence of moderate–severe early-onset OHSS and lowered total estradiol exposure [59,60]. A recent randomized trial in PCOS patients at high risk for hyperstimulation confirmed that adding letrozole did not change the overall occurrence of OHSS but did attenuate its clinical severity [61]. Specifically, letrozole reduced peak E2 levels, decreased the number of oocytes retrieved, and thereby shifted any OHSS cases toward milder grades [59,61].

Crucially, these estradiol-sparing effects were achieved without compromising key efficacy outcomes—pregnancy rates, endometrial thickness, embryo yield, and live-birth rates remained equivalent to standard COS protocols [62,63]. A 2020 systematic review further showed that letrozole co-treatment across diverse stimulation protocols consistently lowered total OHSS risk compared with control regimens, while miscarriage and multiple-pregnancy rates were unchanged [64].

Thus, although careful cycle monitoring is still mandatory, letrozole—used either as a sole oral agent for anovulatory infertility or as an adjuvant in high-responder IVF cycles—provides an effective strategy to temper OHSS risk. Together with its very low multiple-pregnancy rate and mild side-effect profile, these data support the view that letrozole is a safe, first-line option for ovulation induction from a maternal standpoint.

## 5. Mechanistic Insights into Potential Teratogenicity

Despite robust clinical data supporting the fetal safety of letrozole, initial concerns regarding its potential teratogenicity stemmed from mechanistic hypotheses related to the critical role of estrogens in early embryonic development. Estrogens are essential regulators of key developmental processes, including tissue differentiation, angiogenesis, and organogenesis during the earliest stages of pregnancy [65,66]. As letrozole transiently suppresses estrogen synthesis during the peri-ovulatory period, theoretical concerns emerged that residual drug presence during the immediate post-conception phase could disrupt estrogen-dependent signaling cascades, thereby adversely affecting embryogenesis.

The theoretical concern that letrozole might exert teratogenic effects has been primarily linked to its potential to cause estrogen deprivation during the critical window of organogenesis. Between weeks 3 and 8 post-fertilization, a finely regulated estrogenic milieu is essential for the normal development of key embryonic structures, including the neural tube, heart, and limb buds [65]. Experimental studies in rodents have shown that aromatase inhibition during this period can impair skeletal and visceral development: for instance, rat models exposed to letrozole during gestation days 6–17 demonstrated dose-dependent increases in post-implantation loss and vertebral anomalies [52]. In humans, analogous concerns have focused on the possibility of ventricular septal defects and limb abnormalities if letrozole remains systemically active beyond ovulation and into the early post-conception period [53]. However, two critical clinical safeguards significantly mitigate this risk. First, timing: letrozole is typically administered between cycle days 3 and 7, and given its terminal half-life of approximately 2–4 days, drug levels are negligible by the time of fertilization (~cycle day 14) [22,32]. Second, dose: ovulation-induction protocols utilize modest doses (2.5–7.5 mg/day) that result in transient, moderate estrogen suppression, in contrast to the prolonged and high-dose exposures associated with teratogenicity in animal studies [52]. As a result, the human embryo is rarely exposed to pharmacologically significant levels of letrozole during organogenesis, and this is supported by epidemiological data showing no increased incidence of congenital anomalies in letrozole-conceived pregnancies [17,67].

Another mechanistic concern related to potential teratogenicity involves placental aromatase suppression during early gestation. The human placenta exhibits increasing aromatase (CYP19A1) activity beginning in the late first trimester, playing a critical role in the local conversion of androgens to estrogens [68]. If letrozole were to persist in the maternal circulation after implantation, it could theoretically reach the fetoplacental unit and interfere with trophoblastic estrogen production, thereby impairing villous vascularization. This possibility is supported by findings from an experimental rat model, in which sustained aromatase inhibition led to the formation of placental cysts and the subsequent development of late-onset intrauterine growth restriction (IUGR) [53]. However, in clinical ovulation induction protocols, letrozole is administered exclusively during the early follicular phase and is typically cleared from the maternal system well before chorionic villi are formed. By the time placental aromatase expression peaks, endogenous estrogen synthesis far exceeds any transient suppression that may have occurred earlier in the cycle. Consistent with this pharmacologic and temporal dissociation, clinical cohort studies have reported no increase in rates of pre-eclampsia, IUGR, or preterm birth in pregnancies conceived using letrozole [67].

A more speculative yet important area of investigation concerns epigenetic and genomic alterations potentially induced by transient hormonal perturbations during gametogenesis or early embryogenesis. Although assisted reproductive technologies (ART) as a whole have been associated with subtle epigenetic changes, such as altered DNA methylation patterns at imprinted loci [69], no study to date has demonstrated letrozole-specific epigenetic abnormalities. Follow-up studies of children conceived following letrozole-based ovulation induction have consistently shown normal growth, neurodevelopment, and health outcomes. Moreover, comparative analyses of placental or cord blood DNA methylation between letrozole-conceived and naturally conceived offspring have not revealed any consistent or clinically meaningful differences [67]. While long-term prospective cohorts will be required to definitively assess whether transient estrogen suppression around the time of conception leaves an epigenomic footprint, the currently available evidence is highly reassuring.

Taken together, these mechanistic pathways—estrogen deprivation during organogenesis, transient placental aromatase inhibition, and theoretical epigenetic modulation—offer plausible explanations for the early safety concerns associated with letrozole use in fertility treatment. However, the convergence of mechanistic safeguards (short duration, early-cycle timing, low dosage) with consistently reassuring clinical data has largely dispelled these concerns, supporting letrozole’s safety profile in ovulation induction.

## 6. Risk Mitigation Strategies and Clinical Recommendations

Optimal safety with letrozole relies on administering the drug only in the early follicular phase, at the lowest effective dose, with appropriate ultrasound/hormonal monitoring and patient-specific protocol adjustments. Contemporary guidelines and multiple clinical trials confirm that, when these principles are followed, letrozole provides excellent efficacy with minimal maternal or fetal risk.

Optimal timing of administration: Letrozole should be started on cycle day 2–3 and limited to a 5 (occasionally 7)-day course so that treatment finishes by about cycle day 7–9; with a terminal plasma half-life of ~45 h, drug levels are negligible by the time of ovulation, thereby minimizing embryonic exposure [22].

Randomized and retrospective studies comparing start days (CD 3 vs. 5) demonstrate equivalent or superior pregnancy rates when letrozole is confined to the earliest follicular days, supporting this timing strategy [70]. Protocols that continue letrozole into, or beyond, the peri-ovulatory period provide no efficacy advantage and should be avoided.

Use the lowest effective dose: Most anovulatory women ovulate on 2.5 mg daily for five days; higher doses (5–7.5 mg) yield more follicles but not consistently higher live-birth rates and may slightly increase multiple-pregnancy risk [71]. Dose-escalation trials (2.5 mg vs. 5 mg vs. 7.5 mg) show diminishing returns beyond 5 mg [72]. Extended 10-day regimens should be reserved for true letrozole-resistant PCOS cases [73].

Ultrasound and hormonal monitoring: Mid-cycle transvaginal ultrasound (≈CD 12) confirms response, helps time intercourse/IUI, and detects rare over-response (>2 mature follicles) that would increase multiple-gestation risk [74]. Follicular monitoring studies correlate a leading-follicle size of 18–28 mm with optimal pregnancy rates while keeping twins uncommon [74]. Daily or alternate-day scans in research settings illustrate that letrozole cycles usually recruit a single dominant follicle and maintain favorable endometrial thickness [75]. Where ultrasound resources are limited, home LH-surge kits plus the lowest effective dose are acceptable compromises.

Personalization of protocol: Personalized protocol adaptation is essential to optimize letrozole efficacy and minimize risks across different clinical subgroups. In women with PCOS, current international guidelines recommend lifestyle optimization—including a modest weight loss of 5–10% in those with a body mass index (BMI) > 25 kg/m^2^—as a first-line strategy to enhance ovulatory and pregnancy outcomes in conjunction with letrozole therapy [6]. In insulin-resistant or obese PCOS phenotypes, the addition of metformin to letrozole has been shown to improve ovulation rates more effectively than combinations such as metformin with clomiphene or surgical interventions like ovarian drilling [76].

For patients with high ovarian reserve or those at elevated risk of hyper-response—such as individuals with numerous antral follicles—treatment should commence with the lowest effective letrozole dose (typically 2.5 mg daily) to minimize the risk of multifollicular development, with cautious escalation only if anovulation persists [20,28]. Conversely, in women with low ovarian reserve or of advanced maternal age, combining letrozole with low-dose gonadotropins offers a “mild stimulation” approach that may increase follicular yield without triggering an exaggerated ovarian response, making it a suitable strategy for those requiring a more tailored regimen [77].

Letrozole also plays a critical role in fertility preservation protocols for patients with estrogen-sensitive malignancies, particularly breast cancer. In this setting, co-treatment protocols incorporating letrozole during controlled ovarian stimulation have been shown to significantly attenuate peak estradiol levels while preserving oocyte yield. Importantly, the use of such protocols in breast cancer survivors has not been associated with an increased risk of cancer recurrence, and is therefore widely adopted in oncofertility practice [78,79].

Counseling and informed consent: Letrozole for infertility remains off-label in many jurisdictions; informed consent should summarize large randomized evidence showing higher live-birth and lower twin rates versus clomiphene (27.5% vs. 19.1% live-birth; 3.4% vs. 7.4% twins) [40] and reassure patients that subsequent meta-analyses find no increase in congenital anomalies or miscarriage [39]. Common transient side effects (fatigue, headache, hot flushes) occur in <10% of cycles [71,72].

Adjunctive precautions: Progesterone luteal support can be considered in women with prior luteal-phase deficiency, though not mandatory for standard oral cycles [75]. Drug-interaction potential is low given the short treatment window, but strong CYP2A6/CYP3A4 inducers or inhibitors should be avoided concurrently [22]. Finally, always exclude an unrecognized early pregnancy before starting letrozole and postpone treatment if significant, uncontrolled comorbidities render pregnancy unsafe.

## 7. Knowledge Gaps and Future Directions

Despite significant advances in our understanding of letrozole’s role in infertility treatment, several knowledge gaps persist and opportunities for further research remain. Large multicenter prospective studies are still needed. Most randomized trials to date focus on PCOS or single-center cohorts; systematic reviews repeatedly note that rare outcomes can only be conclusively assessed by much larger registries [17]. A prospective international registry enrolling women treated with letrozole for PCOS, unexplained infertility and other indications would allow detection of uncommon events (e.g., specific birth defects occurring at <1:10,000) and improve generalizability across ethnic groups [80].

Long-term offspring follow-up is limited. Nearly all studies stop at birth or early infancy. The only published pregnancy-registry follow-up to 3 years reported normal growth and cognitive development in letrozole-conceived children versus clomiphene/gonadotropin controls, but adolescence and adult reproductive health remain unstudied [81]. Setting up longitudinal cohorts will clarify puberty timing, metabolic profiles and eventual fertility in this population.

Epigenetic and genomic effects require deeper exploration. Reviews of ART highlight potential imprinting and DNA-methylation changes after assisted conception, yet letrozole-specific data are scarce [69]. Targeted cord-blood and placental studies could examine methylation of estrogen-responsive or imprinted loci in letrozole conceptions. Pharmacogenomic work shows CYP2A6 polymorphisms alter letrozole clearance and might prolong peri-conception drug exposure [82]; studying whether such variants modulate fetal epigenetic signatures is a logical next step.

Real-world data and registries can complement RCTs. Retrospective population-based cohorts from China and Europe already suggest no excess neonatal morbidity with letrozole [19], but continuous surveillance via a global “Letrozole OI Registry” would capture practice patterns and outcomes in less-studied groups such as hypothalamic amenorrhea or advanced maternal age. Such databases also enable cost-effectiveness analyses and machine-learning models to predict response.

Unanswered clinical questions persist. For example:-Managing non-responders: Extended 7- or 10- day regimens rescue ovulation in up to 90% of letrozole-resistant PCOS patients, but comparative trials versus immediate gonadotropin step-up are lacking [73].-Dose and duration optimization: Recent comparative work in 2024 confirmed better follicular recruitment with higher starting doses or extended courses, but robust live-birth data are still lacking [30].-Upper age limits: Evidence is sparse on whether women ≥ 40 years benefit meaningfully from oral letrozole versus expedited IVF; pragmatic trials stratified by age and ovarian reserve could guide practice.-Combination regimens: Meta-analyses call for studies on low-dose letrozole combined with other oral agents or mild FSH to enhance unexplained infertility outcomes without increasing multiples [62].-Patient-reported outcomes: Quality-of-life and treatment-burden data comparing letrozole with clomiphene or injectables are scarce; dedicated PRO instruments should be incorporated into future trials [83].

## 8. Conclusions

The evidence reviewed confirms that letrozole occupies a uniquely favorable position at the intersection of clinical efficacy and fetal–maternal safety for ovulation induction. Robust randomized trials demonstrate that, compared with clomiphene citrate, letrozole yields higher ovulation, pregnancy, and live-birth rates—particularly in anovulatory polycystic ovary syndrome—while at the same time reducing the incidence of multiple gestation and avoiding the anti-estrogenic endometrial effects seen with selective estrogen-receptor modulators. Mechanistically, its brief, early-cycle aromatase inhibition induces a physiologic, monofollicular response and is largely cleared before the critical window of human organogenesis, a pharmacokinetic profile that underpins its reassuring fetal safety record. Extensive meta-analyses and large cohort studies now show no increase in congenital anomalies, miscarriage, or adverse perinatal outcomes relative to alternative therapies or natural conception, and maternal adverse events are generally mild and transient. When used according to guideline-endorsed principles—early follicular timing, lowest effective dose, ultrasound monitoring, and individualized protocols—letrozole offers a highly favorable benefit–risk profile across diverse infertility scenarios, including unexplained infertility, mild stimulation protocols, and fertility preservation for estrogen-sensitive cancers.

Continued vigilance remains essential. Large multicenter registries and long-term follow-up of offspring will further clarify rare outcomes, potential epigenetic effects, and optimal management of letrozole-resistant cases, while real-world data can refine cost-effectiveness and patient-reported experience. Nonetheless, the current totality of evidence supports letrozole as the preferred first-line oral agent for ovulation induction and as a versatile adjunct in assisted-reproduction protocols, aligning effective fertility care with the paramount goals of maternal and fetal safety.

## Figures and Tables

**Figure 1 biomedicines-13-02051-f001:**
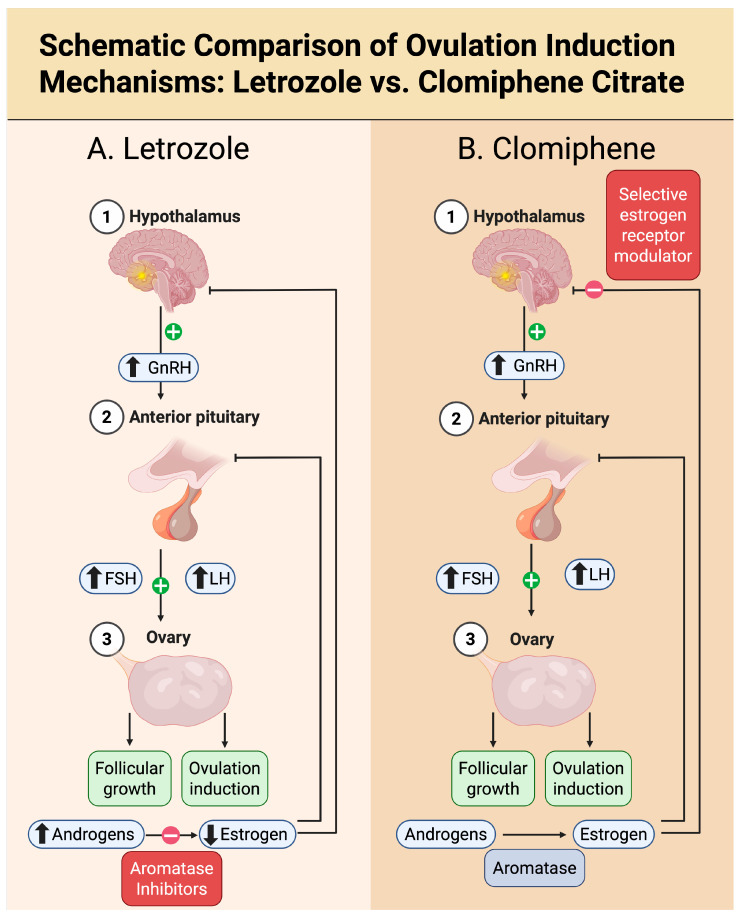
Mechanisms of ovulation induction with letrozole versus clomiphene citrate. (**A**). Letrozole, a nonsteroidal aromatase inhibitor, suppresses estrogen synthesis in the ovary, leading to reduced circulating estradiol levels (↓ estrogen). This relieves negative feedback on the hypothalamus and anterior pituitary, resulting in increased GnRH secretion and enhanced release of FSH and LH (↑ GnRH, FSH, LH). The rise in FSH stimulates follicular growth and ovulation. Letrozole does not block estrogen receptors, thereby preserving normal estrogen signaling in peripheral tissues. (**B**). Clomiphene citrate, a selective estrogen receptor modulator, acts centrally by binding to hypothalamic estrogen receptors, blocking endogenous estradiol’s negative feedback (dashed red line). This mimics a hypoestrogenic state, increasing GnRH secretion and stimulating pituitary release of FSH and LH. The prolonged anti-estrogenic activity of clomiphene may lead to the development of multiple follicles and can adversely affect estrogen-sensitive tissues such as the endometrium and cervical mucus. Created in BioRender. Kaltsas, A. (2025) https://BioRender.com/57b481f.

**Table 1 biomedicines-13-02051-t001:** Comparison of ovulation induction agents (letrozole vs. clomiphene vs. gonadotropins vs. gonadotropins (FSH ± LH)).

Aspect	Letrozole (Aromatase Inhibitor)	Clomiphene Citrate (SERM)	Gonadotropins (FSH ± LH)
Mechanism	Inhibits aromatase → lowers estrogen → releases GnRH/FSH from negative feedback. Stimulates endogenous FSH surge for follicle development.	Blocks estrogen receptors in hypothalamus/pituitary → perceived hypoestrogenic state → increased GnRH/FSH. Also binds peripheral receptors (anti-estrogen effect on endometrium, cervix).	Direct ovarian stimulation with exogenous FSH (±LH), bypassing the hypothalamus. Higher likelihood of multifollicular recruitment without careful dose titration.
Typical Regimen	An amount of 2.5–7.5 mg orally and daily for 5 days (e.g., CD 3–7). Can extend to 7–10 days if no response. Off-label use for OI.	An amount of 50–150 mg oral daily for 5 days (CD 3–7). Max ~6 cycles recommended. FDA-approved for OI.	Daily subcutaneous injections (~75–150 IU FSH) for ~10–14 days with ultrasound monitoring. Often combined with hCG trigger; used with IUI/IVF.
Ovulation Rate	In total, ~60–80% in anovulatory PCOS (higher than clomiphene in RCTs). Many CC-resistant patients will ovulate on letrozole.	In total, ~50–70% in anovulatory PCOS (lower in clomiphene-resistant cases). Diminished efficacy in obese PCOS.	In total, >80–90% with adequate dosing (nearly all will ovulate given sufficient stimulation). Risk of multifollicular ovulation rises with higher doses.
Live Birth/Success	Higher cumulative live birth rate in PCOS vs. clomiphene (27.5% vs. 19.1% over up to 5 cycles). Comparable pregnancy rates to clomiphene in unexplained infertility. Often used before moving to IVF.	Proven efficacy for decades; ~15–20% live birth over ~6 cycles in PCOS, but inferior to letrozole in recent trials. Historically first-line in unexplained infertility (now often replaced by letrozole).	Per-cycle live-birth is context-dependent:• IUI: ~10–15%• Controlled ovarian stimulation + IUI (unexplained): up to ~30%• IVF: live-birth per transfer ~30–50%; risk of multiples mitigated with single-embryo transfer.
Multiple Pregnancy	In total, ~3–5% (mostly twins). Lowest risk among OI agents due to monofollicular tendency. High-order multiples (≥triplets) are rare.	In total, ~5–8% (mostly twins). Higher than letrozole, as clomiphene often yields 2–3 follicles. Triplets uncommon but have been reported.	Highest if multiple follicles are inseminated:• IUI twin rates ~20% when many follicles develop• Significant risk of higher-order multiples without careful cancelation criteria• In IVF, risk is controllable via single-embryo transfer.
Endometrial Effect	Neutral or positive—does not deplete estrogen receptors. Endometrium usually remains receptive (adequate thickness). Implantation rates in letrozole cycles are favorable.	Negative—estrogen receptor down-regulation can thin endometrium and reduce cervical mucus quality. This can lower implantation despite ovulation success.	Typically positive (supraphysiologic E2 thickens endometrium); however, very high E2 (e.g., in OHSS) may impair receptivity; luteal support often required if GnRH-agonist trigger is used.
Side Effects	Mild and related to estrogen drop: reversible fatigue, dizziness, headaches in ~5–10%. Hot flashes less common.	Menopausal-type side effects: hot flashes, night sweats, mood swings, breast tenderness. In total, ~1–2% report visual blurring or scotomata.	Bloating, injection-site discomfort, mood changes. Primary concern: OHSS in over-responders; requires preventive monitoring and cycle adjustments.
Monitoring and Cost	Ultrasound monitoring recommended for safety, but in low-resource settings sometimes minimal monitoring is performed. Very inexpensive per pill.	Ultrasound monitoring improves safety (to check follicle number/size). Low cost per pill.	High medication and monitoring costs. Requires intensive ultrasound ± labs for dose titration and safety (OHSS/multiples risk management).

Abbreviations: FSH = follicle-stimulating hormone; LH = luteinizing hormone; GnRH = gonadotropin-releasing hormone; CD = cycle day; OI = ovulation induction; IUI = intrauterine insemination; IVF = in vitro fertilization; PCOS = polycystic ovary syndrome; OHSS = ovarian hyperstimulation syndrome; FDA = U.S. Food and Drug Administration; hCG = human chorionic gonadotropin; RCT = randomized controlled trial; E2 = estradiol. Arrows (→) denote the sequence of biological events or consequences (“leads to”/“results in”).

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
