# Peer review of "Letrozole at the Crossroads of Efficacy and Fetal Safety in Ovulation Induction: A Narrative Review"

_biomedicines, 2025, doi:10.3390/biomedicines13092051_

Round 1
Reviewer 1 Report
Comments and Suggestions for Authors
This review provides an in-depth analysis of the scientific evidence on the efficacy of letrozole as an ovulation inducer, as well as evidence on its safety for the fetus.
The introduction contains all the necessary elements to introduce the main topic, which is ovulatory disorders and, therefore, fertility due to the presence of PCOS. It then presents the evidence related to the positioning of letrozole as an ovulation inducer, despite being a drug originally designed for the treatment of breast cancer. At the same time, it compares its efficacy with other drugs used as ovulation inducers, such as clomiphene. The objective of this review is perfectly stated, since the manuscript carries out a careful evaluation of the efficacy of letrozole as an ovulation inducer, its safety in the event of possible fetal exposure, and its application in reproductive medicine. This objective is achieved through the systematic and concise description of relevant works regarding the mechanisms involved in the action of letrozole, its efficacy, and its safety in the event of possible fetal exposure.
The description of the pharmacological mechanism of action compared to other ovulation-inducing drugs is well presented, supported by appropriate bibliography and accompanied by pertinent graphics. It then reviews the pharmacokinetics, giving the reader a detailed summary of its administration, which later helps explain why its administration is safe and does not overlap with fetal development.
The review then provides a detailed comparison with other drugs used for ovulation induction, such as clomiphene and gonadotropins. The text is accompanied by a well-designed and comprehensive comparison table. This comparison provides well-documented evidence on the effectiveness of letrozole treatment for ovulation induction. The review continues with a section on the safety of letrozole administration for both the mother and the fetus. It provides data from the perspective of preclinical research evidence in animal models, regarding the safety of possible congenital effects, and then on the development of the fetus and newborn, as well as the possible consequences of its administration for the mother. Towards the end, it addresses possible teratogenic consequences and compares them with well-documented evidence regarding the same consequences in pregnancies without letrozole treatment. To conclude, the review addresses the evidence related to the recommendations and precautions for the administration of letrozole treatment. Finally, it provides a concise and well-structured description regarding the need to delve deeper into: how to manage non-response to treatment, treatment optimization, age limits for its administration and combination of treatments, as well as the possible epigenetic effects of letrozole administration, for which there is still no conclusive evidence. The work concludes with a clear and conclusive message regarding the use of letrozole for ovulation induction, its comparison with other treatments, and its efficacy and safety. This message is fully aligned with the objective stated at the beginning of the text.
I must congratulate the authors, because it was a work that I enjoyed reviewing.
I just want to make my minor suggestions to improve the quality of the text.
- Figure 1 page 3. The letter within the graph should be larger so that it can be seen at a glance without having to zoom in on it.
- Table 1, pages 5 and 6. The last column regarding gonadotropins doesn't allow for the differentiation of the content related to each row in the table. I recommend reviewing the format to improve its quality.
Author Response
Response to Reviewer 1
We sincerely thank the Reviewer for the thoughtful and encouraging assessment of our review. We appreciate the two specific suggestions regarding Figure 1 and Table 1 and have implemented both to improve readability and clarity.
Comment 1 (Figure 1, page 3):
“The letter within the graph should be larger so that it can be seen at a glance without having to zoom in on it.”
Response: We agree. We have replaced Figure 1 with an updated, high‑resolution version to improve legibility at 100% zoom. Specifically, we increased all label/font sizes by ~30%, enlarged symbols, and modestly increased line weights for better contrast. The figure content and caption remain scientifically unchanged.
Comment 2 (Table 1, pages 5–6):
“The last column regarding gonadotropins doesn’t allow for the differentiation of the content related to each row in the table. I recommend reviewing the format to improve its quality.”
Response: We agree and have reformatted the Gonadotropins column to ensure clear, row‑by‑row differentiation. We clarified the header, broke dense entries into short, sentence‑style lines, and aligned phrasing tightly to each row heading. Representative edits are shown below in bold (full table updated on pp. 5–6).
Header (Table 1):
-
Gonadotropins (FSH ±â€¯LH)
Mechanism row (Gonadotropins column):
-
Direct ovarian stimulation with exogenous FSH (±â€¯LH), bypassing the hypothalamus.
-
Higher likelihood of multifollicular recruitment without careful dose titration.
Typical Regimen row:
-
Daily subcutaneous injections (≈ 75–150 IU FSH) for ~10–14 days with ultrasound monitoring.
-
Often combined with an hCG trigger; used with IUI/IVF protocols.
Ovulation Rate row:
-
> 80–90% with adequate dosing (nearly all will ovulate given sufficient stimulation).
-
Risk of multifollicular ovulation rises with higher doses.
Live Birth / Success row:
Per‑cycle live birth is context‑dependent:
-
IUI: ~10–15%
-
Controlled ovarian stimulation + IUI (unexplained): up to ~30%
-
IVF: live‑birth per transfer ~30–50%; risk of multiples mitigated with single‑embryo transfer.
Multiple Pregnancy Risk row:
-
Highest if multiple follicles are inseminated:
-
IUI twin rates ~20% when many follicles develop
-
Significant risk of higher‑order multiples without strict cancellation criteria
-
In IVF, risk is controllable via single‑embryo transfer.
-
Endometrial Effect row:
-
Typically positive (supraphysiologic E2 thickens endometrium);
-
however, very high E2 (e.g., in OHSS) may impair receptivity; luteal support often required if a GnRH‑agonist trigger is used.
Side Effects row:
-
Bloating, injection‑site discomfort, mood changes.
-
Primary concern: OHSS in over‑responders; requires preventive monitoring and cycle adjustments.
Monitoring and Cost row:
-
High medication and monitoring costs.
-
Requires intensive ultrasound ±â€¯labs for dose titration and safety (OHSS/multiples risk management).
Closing:
We are grateful for the Reviewer’s constructive suggestions. The updated Figure 1 enhances at‑a‑glance readability, and the revised Table 1 now clearly differentiates the gonadotropin content across all rows. We believe these refinements improve the manuscript’s clarity and practical utility for readers.
Reviewer 2 Report
Comments and Suggestions for Authors
Interesting and relevant review, especially since letrozole is still forbidden in some countries due to concerns about fetal safety based on local regulations. This is a very well conducted review and I have no relevant comments. I thank the authors for their efforts to bring up this important topic. In my humble opinion, the review provides a good rationale in the Introduction Section. All relevant data and findings are summarized concisely. Thus, the review provides readers with a good and clinically relevant overview. I recommend to accept the article after minor revisions.
According to my literature research, there is no review of the topic as a whole.
The conclusions are consistent with the evidence and arguments presented.
No additional comments on the tables and figures.
Minor comments:
- Introduction: Letrozole is also recommended by the international PCOS network (Teede et al 2023), which could be worth mentioning in the Introduction Section
- Chapter 2.2: “[…] commonly initiated between cycle day (CD) 3 and CD 5 of a spontaneous or progesterone-induced menses.” – Could be rephrased as “[…] commonly initiated from cycle day (CD) 3 to CD 5 of a spontaneous or progesterone-induced menses.”
Author Response
Response to Reviewer 2
We sincerely thank the Reviewer for the supportive and encouraging evaluation of our work. We appreciate the two minor suggestions and have implemented both.
Comment 1 (Introduction):
“Letrozole is also recommended by the international PCOS network (Teede et al., 2023), which could be worth mentioning in the Introduction Section.”
Response: Thank you. We have added an explicit statement in the Introduction acknowledging the international PCOS network guideline (Teede et al., 2023), harmonized with our existing guideline citations.
Change in manuscript (Introduction, page 2):
-
Inserted sentence: “Additionally, the international evidence‑based guideline spearheaded by the international PCOS network also endorses letrozole as the first‑line ovulation induction therapy for anovulatory PCOS.”
Comment 2 (Section 2.2 phrasing):
Rephrase “… commonly initiated between cycle day (CD) 3 and CD 5 …” to “… commonly initiated from cycle day (CD) 3 to CD 5 …”
Response: Implemented as suggested for clarity and style consistency.
Change in manuscript (Section 2.2, Pharmacokinetics and Dosing for Infertility, page 4):
Original: “Letrozole is administered orally at 2.5 mg daily for five days, commonly initiated between cycle day (CD) 3 and CD 5 of a spontaneous or progesterone‑induced menses.”
Revised: “Letrozole is administered orally at 2.5 mg daily for five days, commonly initiated from cycle day (CD) 3 to CD 5 of a spontaneous or progesterone‑induced menses.”